# BrainAE: Alignment-driven Autoencoder for Bidirectional Visual Encoding and Decoding

## Abstract

Modeling the bidirectional mapping between visual stimuli and neural activity is critical for both neuroscience and brain–computer interfaces (BCIs). Although significant progress has been made in independently addressing visual encoding and decoding, **unified latent representations for the bidirectional mapping remain lacking**. Here, we propose **BrainAE**, an autoencoder-based framework designed for both visual encoding and decoding. Contrastive alignment with image models drives the latent features **toward a shared representation space of visual stimuli and neural responses**. Once trained, the model supports **stimulus-to-brain encoding**, **brain-to-stimulus decoding**, and **whole-brain signal reconstruction**. We extensively evaluate the model on electrophysiology, including human electroencephalography (EEG) and magnetoencephalography (MEG), as well as macaque multi-unit spiking activity (MUA), spanning non-invasive and invasive recordings, macro- and micro-scales, and species. Results demonstrate competitive encoding and decoding performance, revealing spatial, temporal, and semantic patterns consistent with established neuroscience findings. BrainAE provides a methodological foundation for probing brain function and developing BCIs.

## 1 Introduction

Human visual systems process visual stimuli into neural representations, enabling perception and decision-making DiCarlo & Cox (2007). Resolving and replicating how our brain represents visual information remains a central challenge in understanding the mechanisms of visual systems Wang et al. (2025); Kay et al. (2008); Kamitani & Tong (2005). Two primary tasks involved in visual modeling aim to bridge the mapping between visual stimuli and brain responses: **visual encoding**, which predicts neural activity elicited by stimuli, and **visual decoding**, which identifies or generates stimuli from neural activity Naselaris et al. (2011); Gao et al. (2021).

Visual encoding has seen progress accompanied by interpretable mechanisms O'Shea et al. (2025). Studies established the primary visual cortex (V1) encoded basic features like edges and orientations Li et al. (2025), while higher-order regions process complex patterns, object concept, and spatial location DiCarlo et al. (2012); Kar et al. (2019). The development of encoding models has also evolved from linear models to advanced nonlinear models, incorporating deep learning techniques Yamins & DiCarlo (2016); Tang et al. (2023). By aligning with Convolutional Neural Networks (CNNs), researchers have explored the hierarchical structure encoding from low- to high-level representations Gu et al. (2022). Recurrent neural networks (RNNs) have also been applied to model dynamic stimuli such as videos and natural scenes, capturing temporal and contextual dependencies in neural activity (Horikawa & Kamitani, 2017; Lahner et al., 2024). Recent data-driven models trained on large-scale datasets have begun to show competitive performance with strong generalization Du et al. (2025). The consistent features between brain and artificial models emphasize the effectiveness of aligning neural and computational representations Shen et al. (2025); Franke et al. (2025).

Visual decoding has also been significantly propelled by brain recordings and machine learning. We can now identify low-level features such as brightness, contrast, and motion Liu et al. (2021); Shi et al. (2024); Song et al. (2024), as well as higher-level semantics Schneider et al. (2023). It provides great temporal and spatial resolution for fast and accurate visual decoding using intracranial signals like local field potentials (LFP) Liu et al. (2009) and multi-unit spiking activity (MUA), and non-invasive signals such as magnetoencephalography (MEG) Cichy et al. (2014); Benchetrit et al. (2023) and low-cost electroencephalography (EEG), which demonstrated potential to expand the applicability of daily brain-computer interfaces (BCIs)Liu et al. (2024). Early machine learning approaches,

including Bayesian models and support vector machines, were instrumental in decoding Wu et al. (2016). More recently, data-driven methods leveraging large-scale data such as THINGS-EEG Gifford et al. (2022), THINGS-MEG Hebart et al. (2023), and the Natural Scenes Dataset (NSD) Allen et al. (2022); Chen et al. (2023); Takagi & Nishimoto (2023) have demonstrated remarkable success. Deep learning architectures, including CNNs, Transformers Azabou et al. (2023), particularly self-supervised learning models Schneider et al. (2023), have been employed to extract brain features aligning with visual features provided by artificial models Song et al. (2023a); Li et al. (2024).

Despite rapid advances, **encoding and decoding are usually developed in isolation, although both rely on shared neural mechanisms**. **Aligning brain and artificial representations may offer a promising approach to bridge bidirectional mapping**, providing both stronger performance and neuroscience insights. Here, we propose Brain Autoencoder (**BrainAE**) to **unify visual encoding and decoding with shared representation space**. The model has an encoder to extract neural activity features and a decoder to reconstruct activity from the latent. In parallel, we leverage pre-trained image models to provide visual features and use contrastive alignment to drive the two kinds of features closer. The unified latent space enables the model to predict neural activity from visual stimuli (encoding), classify and generate visual stimuli based on brain activity (decoding), and reconstruct neural activity from masked recordings (reconstruction). We validate BrainAE using **datasets with high time resolution and diverse spatial scales**, including non-invasive EEG and MEG and invasive MUA. The results show that BrainAE effectively **simulates brain activity while preserving characteristic neural patterns**. It **achieves strong decoding performance and discovers critical spatial and temporal regions** involved in visual processing. Moreover, the framework **reveals consistency between encoding and decoding** on spatial and temporal dimensions.

Our main contributions are summarized as follows:

- Introducing an autoencoder framework that aligns brain and artificial representations within a bidirectional space, enabling **brain function simulation (encoding)**, **brain information extraction (decoding)**, and **whole-brain activity prediction (reconstruction)**.

- Achieving strong performance and neuroscientific plausibility, along with high encoding correlation, preserving original brain dynamics, and superior decoding accuracy and stimulus generation, discovering meaningful spatial, temporal and semantic patterns.

- Demonstrating broad generalization with EEG, MEG, and MUA, covering diverse recordings (non-invasive and invasive), scales (macro and micro), and species (human and macaque).

## 2 RELATED WORKS

Aligning neural activity with artificial models has become an attractive goal in computational neuroscience. Goal-driven hierarchical CNNs were introduced to model neural responses in higher visual areas by mapping stimuli to brain activity Yamins & DiCarlo (2016). CORnet extended the model to capture the object recognition mechanisms of the brain Kubilius et al. (2019), and further work emphasized the necessity of recurrent processing to model the ventral stream Kietzmann et al. (2019); Kar et al. (2019). More complex architectures have since been explored, including task-optimized CNNs with recurrent gating Nayebi et al. (2022) and non-standard architectures for modeling the mouse vision Conwell et al. (2021). Other studies have highlighted a divergence between artificial and biological vision, showing that task-optimized deep networks may not align well with inferotemporal (IT) cortex representations Linsley et al. (2023). Encoder–decoder models have also been applied to visual coding with fMRI data Han et al. (2019); Qian et al. (2024). Together, these works underscore the importance of high-quality representations in modeling neural populations.

Alignment has also proven effective for decoding brain activity. Latent diffusion models have been used to reconstruct images from brain imaging Takagi & Nishimoto (2023), while contrastive learning with diffusion priors has enabled image retrieval and reconstruction Scotti et al. (2023). Adversarially guided alignment has achieved high-fidelity video reconstruction Chen et al. (2023). Time-resolved EEG and MEG show good potential and feasibility of decoding visual representations with alignment models, revealing key patterns of visual processing Song et al. (2023a); Benchetrit et al. (2023). Parallel advances in latent modeling have also enriched decoding: Latent Factor Analysis via Dynamical Systems (LFADS) was developed for single-trial motor prediction Pandarinath et al. (2018) and later extended with AutoLFADS to improve generalization across brain areas and tasks Keshtkaran

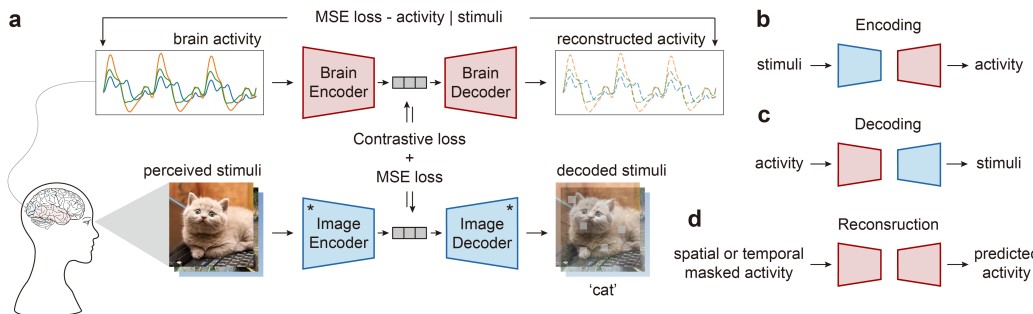

Figure 1: (a) BrainAE diagram. Both the brain activity and the perceived stimuli module contain an encoder and a decoder, where contrastive and MSE loss shape the latent features. MSE loss is also applied between the reconstructed activity and the raw activity. (b) The encoding task predicts brain activity from image stimuli. (c) The decoding task identifies and generates the image stimuli from brain activity. (d) The reconstruction task predicts the complete raw activity from masked activity.

et al. (2022), while tokenization and cross-attention have been introduced as a general decoding framework for large-scale neural recordings Azabou et al. (2023).

While encoding and decoding research have yielded high-quality brain representations, these approaches remain commonly separate and unidirectional. BrainAE bridges this gap by integrating encoding and decoding within a unified framework, especially leveraging alignment with large-scale artificial models to enhance the latent representations for both visual stimuli and brain responses.

## 3 METHODS

### 3.1 OVERVIEW

The overall architecture is illustrated in Fig. 1, comprising two primary modules: the brain module and the image module, both equipped with an encoder and a decoder. The brain encoder maps neural activity to an embedding, while the image encoder transforms visual stimuli into corresponding embeddings. These embeddings serve three purposes: i) visual encoding: image embeddings are passed to the brain decoder to predict neural activity associated with specific visual stimuli. ii) visual decoding: brain embeddings are used to identify or generate visual stimuli via either a template-matching approach or a generative model-based image decoder. Besides, we evaluate the model's capability by reconstructing the raw brain activity from spatially or temporally masked recordings.

### 3.2 PROBLEM DEFINITION

The BrainAE framework consists of a brain encoder $\mathcal{E}_B$, brain decoder $\mathcal{D}_B$, image encoder $\mathcal{E}_I$, and image decoder $\mathcal{D}_I$, as depicted in Fig. 1(a). The model takes as inputs perceived image stimuli $\boldsymbol{X}_i^I$ and the corresponding brain activity signals $\boldsymbol{X}_i^B \in \mathbb{R}^{C \times T}$, where $C$ denotes the number of channels, and $T$ represents the time samples. The $\mathcal{E}_B$ and $\mathcal{E}_I$ transfer the $\boldsymbol{X}_i^B$ and $\boldsymbol{X}_i^I$ into embeddings, $\boldsymbol{z}_i^B \in \mathbb{R}^F$ and $\boldsymbol{z}_i^I \in \mathbb{R}^F$, where $F$ is the embedding size. Then the $\boldsymbol{z}_i^B$ and $\boldsymbol{z}_i^I$ are reconstruct towards the raw brain activity $\boldsymbol{X}_i^{B'}$ and stimuli $\boldsymbol{X}_i^{I'}$ by $\mathcal{D}_B$ and $\mathcal{D}_I$, respectively.

We evaluated the framework with three tasks: Visual encoding, in Fig. 1(b), gets image embeddings $\boldsymbol{z}^I$ by the encoder $\mathcal{E}_I$ and input to the decoder $\mathcal{D}_B$ for the encoded brain activity. Visual decoding, in Fig. 1(c), extracts brain embeddings $\boldsymbol{z}^B$ by the encoder $\mathcal{E}_B$. Then we perform identification, including classification and retrieval, by matching the templates prepared with images belonging to the test condition, before the inference stage. Templates for classification are constructed with several images that never appeared as visual stimuli, while the templates for retrieval use specific visual stimulus images. We implement visual generation leveraging generative models as the decoder $\mathcal{D}_I$.

In addition to primary visual encoding and decoding, we set a new task by adding a mask in spatial and temporal dimensions and reconstruct the raw brain activity with $\mathcal{E}_B$ and $\mathcal{D}_B$, as shown in Fig. 1(d). This task assesses the framework's ability to handle neural activity even with incomplete data.

### 3.3 NETWORK ARCHITECTURE

#### 3.3.1 BRAIN MODULE

The brain module sets up a concise temporal-spatial convolution (TSConv) encoder to extract features from raw neural activity, which are band-pass filtered and standardized. The encoder begins with a 1D convolutional layer that captures temporal features using $k$ kernels of size $(1, m_1)$ and stride of $(1, 1)$. An average pooling layer with a kernel of $(1, m_2)$ and a stride of $(1, s)$ is introduced to alleviate overfitting. Next, spatial features are captured with another 1D convolutional layer using $k$ kernels of size $(ch, 1)$ and stride of $(1, 1)$, where $ch$ usually equals $C$. Convolutional layers are followed by batch normalization and exponential linear unit (ELU) activation for stability and nonlinearity Clevert et al. (2016). Finally, a linear layer transforms the extracted features into a latent space compatible with the image module, ensuring alignment between the two branches of the framework. The brain decoder mirrors the encoder, replacing convolutional and pooling layers with transposed convolution and up-sampling layers to reconstruct brain activity. Details in Appendix D.1.

#### 3.3.2 STIMULI MODULE

Popular image encoders were employed to process perceived stimuli. Encoders pre-trained on large-scale image datasets give us a larger sample space, thus helping generalization. Several models, including Vision Transformer (ViT) Dosovitskiy et al. (2021), Contrastive Language-Image Pre-training (CLIP) Radford et al. (2021), and EVA-CLIP Sun et al. (2023) are involved for demonstration in this work. After the frozen image encoder, we add a linear layer to project features into the shared space. For the classification task, we construct templates with several images belonging to the test condition but not appearing as the visual stimuli, while we directly use the test stimuli as templates for the retrieval task. We test image generation with an image decoder based on prior and pre-trained diffusion models, referring to Li et al. (2024); Scotti et al. (2023). Details in Appendix D.2.

### 3.4 OBJECTIVE FUNCTIONS

#### 3.4.1 CONTRASTIVE LOSS

The training algorithm is given in Appendix E. To align visual stimuli and brain activity in the shared space, we employ contrastive learning based on the InfoNCE loss van den Oord et al. (2019) as:

$$\mathcal{L}_{con} = -\frac{1}{N} \sum_{i=1}^{N} \log \frac{e^{f(\boldsymbol{z}_i^B, \boldsymbol{z}_i^I)/\tau}}{\sum_{j=1}^{N} e^{f(\boldsymbol{z}_i^B, \boldsymbol{z}_j^I)/\tau}} \tag{1}$$

where $N$ represents batch size, $\boldsymbol{z}_i^B$ and $\boldsymbol{z}_i^I$ denotes the features of $i$-th brain activity and stimulus image, $f()$ denotes cosine similarity, and $\tau$ is a temperature parameter to control the distribution.

#### 3.4.2 MSE LOSS

We also introduce MSE loss to constrain the embeddings of visual stimuli and brain activity as below:

$$\mathcal{L}_{fea} = \frac{1}{N} \sum_{i=1}^{N} (\boldsymbol{z}_i^B - \boldsymbol{z}_i^I)^2 \tag{2}$$

Besides, we leverage MSE loss to constrain the brain activity reconstructed with brain embeddings:

$$\mathcal{L}_{reconB} = \frac{1}{N} \sum_{i=1}^{N} (\boldsymbol{X}_i^B - \boldsymbol{X}_i^{B'B})^2 \tag{3}$$

as well as the constraint between brain activity encoded from image embeddings:

$$\mathcal{L}_{reconI} = \frac{1}{N} \sum_{i=1}^{N} (\boldsymbol{X}_i^B - \boldsymbol{X}_i^{B'I})^2 \tag{4}$$

where $\boldsymbol{X}_i^{B'B}$, $\boldsymbol{X}_i^{B'I}$ denote the brain activity obtained from brain embeddings and visual embeddings, using the same brain decoder. Therefore, we set the total reconstruction loss $\mathcal{L}_{recon}$:

$$\mathcal{L}_{recon} = \mathcal{L}_{reconB} + \mathcal{L}_{reconI} \tag{5}$$

## 4 RESULTS

### 4.1 DATASETS AND PREPROCESSING

The THINGS-EEG2 dataset includes 10 participants engaged in a rapid serial visual presentation (RSVP) task using images from the THINGS image set Gifford et al. (2022); Hebart et al. (2019). The training phase comprises 1,654 concepts, each represented by 10 images, repeated 4 times. The testing phase includes 200 concepts, each represented by 1 image, repeated 80 times. Stimuli were presented for 100 ms with a 100-ms blank screen, yielding a stimulus onset asynchrony (SOA) of 200 ms. EEG data were epoched from 0 to 1000 ms post-stimulus, filtered to 0.1–100 Hz, and baseline-corrected using the 200-ms pre-stimulus mean. After down-sampling to 250 Hz, data from all 63 electrodes underwent multivariate noise normalization Guggenmos et al. (2018). Repetitions were averaged per image to enhance the signal-to-noise ratio.

The THINGS-MEG dataset includes 4 participants exposed to stimuli with a jittered SOA of 1500 ± 200 ms, consisting of 500-ms image presentations followed by a blank screen Hebart et al. (2023). The training set covers 1,854 concepts with 12 images per concept, while the testing set includes 200 concepts with 1 image per concept (12 repetitions). Zero-shot setting is ensured by excluding test concepts from training. MEG signals were epoched from 0 to 1000 ms post-stimulus, filtered to 0.1–100 Hz, baseline-corrected, and down-sampled to 200 Hz across 271 channels.

The THINGS ventral stream spiking dataset (TVSD) contains MUA recorded from V1, V4, and IT regions of two macaques viewing natural images Papale et al. (2025). The training set includes 1,854 concepts with 12 images each, while the test set contains 100 concepts with 1 image per concept (30 repetitions). The test concepts are excluded from the training. Spike data were epoched with 0-200 ms post-stimulus, baseline-corrected with the -100 ms pre-stimulus, down-sampled to 1000 Hz across all 1024 channels, and averaged by repetitions. Details in Appendix F.

### 4.2 EXPERIMENT DETAILS

Our method was implemented using PyTorch in Python 3.10 and executed on a GeForce 4090 GPU. Model training required nearly 7 minutes, with inference for each trial taking less than 1 millisecond. For each run, 740 trials were randomly selected from the training data as the validation set. The best-performing models were saved based on the minimum validation loss during training, which ran for 50 epochs to ensure convergence. We perform testing once after training, using several unseen images from the THINGS image set as templates for classification. The intermediate stimulus features were obtained before the training stage with pre-trained and frozen image models.

The hyperparameters were set as follows: $k = 40$, $m_1 = 26$, $m_2 = 5$, $s = 5$, and $\tau = 0.07$ (compared in Appendix I). The model training used the Adam optimizer with a batch size of 800, learning rate of 0.001, $\beta_1 = 0.5$, and $\beta_2 = 0.999$, based on previous studies and preliminary experiments. Wilcoxon Signed-Rank test was employed to evaluate significance levels with p-values. Note that statistical analysis was not performed on MEG and Spike datasets due to the limited number of subjects.

### 4.3 ENCODING

#### 4.3.1 OVERALL QUALITY

The encoding task is to predict brain activity by perceiving visual stimuli. We achieved good performance in various metrics as Table 1 and subject results in Appendix G. All datasets of different recordings with varying numbers of channels were involved in the evaluation. Besides, we chose visual areas, with occipital, temporal, and parietal channels, in EEG and MEG data for testing, where inferior temporal (IT) channels were used in MUA data. Pearson's correlation (r) was employed as the primary metric, accompanied by the mean squared error (MSE) for encoding quality.

The alignment brought significant improvement under the commonly linear and nonlinear brain decoder Gifford et al. (2022); Yamins & DiCarlo (2016). For EEG data, alignment significantly improved the linear model with 0.042 ($p < 0.01$) and 0.038 ($p < 0.01$) increments of Pearson's r in all channels and visual channels, separately. The nonlinear model showed higher potential with the help of alignment. The MEG and the MUA datasets show similar trends in using alignment. From another view, the encoding performance on visual channels was significantly higher than on all channels ($p < 0.01$), implying that the model prioritizes task-related features associated with visual perception.

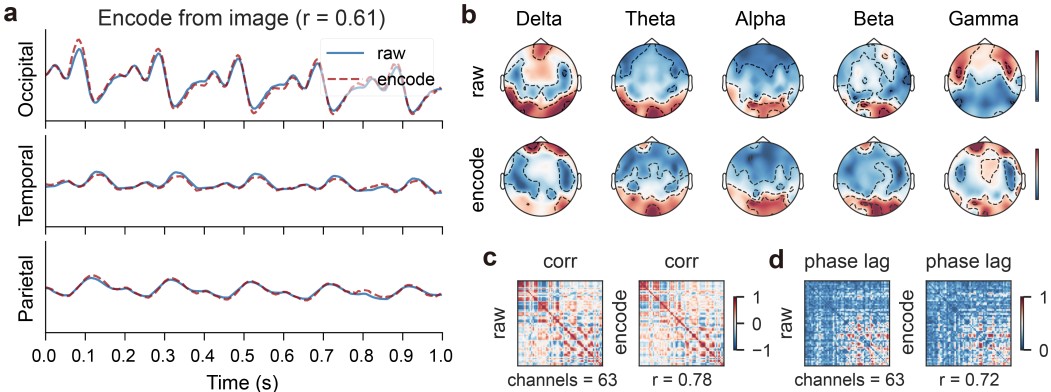

Figure 2: Encoded brain activity and raw activity comparison of one subject in time, space, and frequency visualization. (a) The time signals with averaged channels in occipital, temporal, and parietal areas, separately. (b) The spatial distribution of power spectral density (PSD) across five rhythms. (c) The correlation between electrode channels. (d) The phase lag across electrode channels.

Table 1: Encoding results with BrainAE (predict activity based on visual stimuli).

| DATASET | CHANNEL | Linear Dec | | Nonlinear Dec | | Linear Dec+**Align** | | Nonlinear Dec+**Align** | |
|---|---|---|---|---|---|---|---|---|---|
| | | MSE ↓ | Pearson's r ↑ | MSE ↓ | Pearson's r ↑ | MSE ↓ | Pearson's r ↑ | MSE ↓ | **Pearson's r** ↑ |
| EEG (N=10) | All (ch=63) | 0.037 | 0.480±0.069 | 0.048 | 0.285±0.092 | 0.031 | 0.522±0.067 | 0.027 | 0.583±0.071 |
| | Visual (ch=34) | 0.033 | 0.590±0.066 | 0.042 | 0.448±0.086 | 0.028 | 0.628±0.056 | 0.024 | **0.680±0.052** |
| MEG (N=4) | All (ch=271) | 0.706 | 0.446±0.075 | 0.713 | 0.390±0.071 | 0.629 | 0.482±0.070 | 0.624 | 0.493±0.073 |
| | Visual (ch=152) | 0.754 | 0.508±0.008 | 0.773 | 0.453±0.087 | 0.678 | 0.539±0.078 | 0.669 | **0.551±0.081** |
| MUA (N=2) | All (ch=1024) | 0.141 | 0.796±0.033 | 0.189 | 0.722±0.035 | 0.199 | 0.837±0.018 | 0.292 | 0.852±0.001 |
| | IT (ch=256|320) | 0.079 | 0.689±0.084 | 0.143 | 0.731±0.015 | 0.080 | 0.868±0.038 | 0.066 | **0.898±0.039** |

   i) High correlations have been achieved across different recordings, where alignment shows significant improvement.

   ii) For encoding comparison, there are linear and nonlinear brain decoders in Table 1, and various image encoders in Table 5.

### 4.3.2 TIME, SPACE, AND FREQUENCY

We directly plot encoded brain activity with EEG data to roughly show the encoding quality from temporal, spatial, and frequency aspects. In Fig. 2(a), we show the raw brain activity in blue and the encoded activity in dashed red by averaging the channels of the occipital, temporal, and parietal areas. Visual event-related information across multiple channels has been predicted with similar amplitude and latency over time. Further, we plot the spectral power distribution with spatial topographies in different frequency rhythms, as shown in Fig. 2(b). The raw and encoded brain activity shows obvious similarities in the spatial patterns, especially with the data of theta, alpha, and beta bands.

Because spatial correlation and connectivity are important factors in analyzing time-resolved brain activity, we show the correlation and phase lag between channels in Fig. 2(c) and (d). The raw and encoded signals are still consistent in both matrices with a high Pearson's r, indicating accurate preservation of spatial and temporal patterns in the encoded signals.

### 4.4 DECODING

#### 4.4.1 OVERALL PERFORMANCE

The decoding task is to identify or generate visual stimuli based on brain activity. We show the performance by comparing with state-of-the-art methods in Table 2 and subject-level results in Appendix H. There are 200-way zero-shot tasks for the EEG and MEG datasets, and 100-way for the MUA dataset, where we used top-1 and top-5 accuracy as the metrics. We set classification using the templates constructed with unseen images belonging to test conditions, and retrieval with the stimulus images. Our model achieves good results on both classification and retrieval across EEG and MEG datasets, outperforming other great works, such as BraVL Du et al. (2023), NICE Song et al. (2023a), ATM Li et al. (2024), and MB2C Wei et al. (2024). The latest MUA data also achieves significantly above-chance results with prominent acquisition resolution. These results underscore

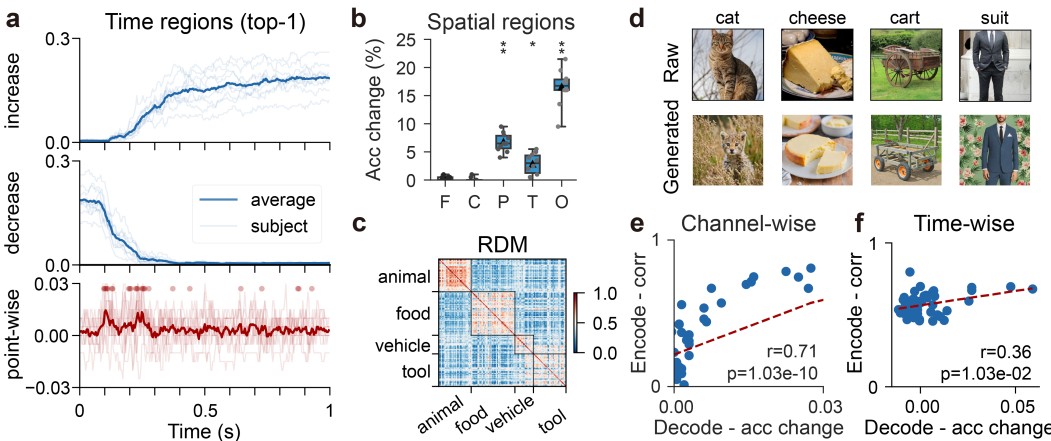

Figure 3: Decoding comparison in temporal, spatial, and semantic aspects. (a) Decoding accuracy changes when different time points of test brain activity are masked. (b) Decoding accuracy changes when different areas of test brain activity are masked. (c) Representational similarity analysis of the brain features to show the semantic information within object categories. (d) The stimuli generated by the features obtained with the trained model. (e) The channel-wise correlation between encoding and decoding results. (f) The time-wise correlation between encoding and decoding results.

Table 2: Decoding results (obtain visual information from activity).

| DATASET | MODEL | Classification | | Retrieval | |
|---|---|---|---|---|---|
| | | Top-1 acc ↑ | Top-5 acc ↑ | Top-1 acc ↑ | Top-5 ↑ |
| EEG (200-way) | BraVL | 5.8±1.3 | 17.5±3.1 | - | - |
| | NICE | 13.8±3.3 | 39.5±6.5 | 18.8±4.9 | 48.0±6.2 |
| | ATM | 6.2±1.5 | 15.4±3.3 | 28.6±6.4 | 58.5±9.0 |
| | MB2C | - | - | 28.5±5.5 | 60.4±6.6 |
| | **BrainAE** | **18.2±3.2** | **46.7±4.3** | **30.6±4.0** | **63.0±4.8** |
| MEG (200-way) | NICE | 10.1±3.5 | 28.4±6.9 | 12.8±3.4 | 36.0±8.1 |
| | ATM | 5.4±3.8 | 15.9±10.2 | 18.4±8.4 | 44.1±14.2 |
| | **BrainAE** | **14.3±4.5** | **35.1±9.7** | **21.4±8.7** | **48.3±13.4** |
| MUA (100-way) | **BrainAE** | **26.9±1.9** | **62.0±3.6** | **43.5±6.1** | **77.0±4.2** |

BrainAE's effectiveness in decoding tasks, highlighting its generalizability across modalities and its capability to bridge brain activity with visual information.

### 4.4.2 TEMPORAL, SPATIAL, AND SEMANTIC ANALYSIS

We analyze the decoding performance with EEG data from different perspectives to illustrate that our feature space is relevant to visual processing in Fig. 3. Here, we first train the overall model and mask different time points of test data in three ways, increasing, decreasing, and point-wise masking along the time, to show the significant response period in Fig. 3(a). From the point-wise results, we can see that losing data between 100-300 ms has a more significant impact on top-1 accuracy ($p < 0.05$), consistent with existing visual processing findings Liu et al. (2009); Xu et al. (2023).

We also show the spatial pattern by masking the channels of different areas in Fig. 3(b). The channels in the occipital ($p < 0.01$), temporal ($p < 0.05$), and parietal ($p < 0.01$) areas, along the ventral and dorsal pathways Bao et al. (2020), show a significant impact on the top-1 accuracy across subjects.

Semantic information is one of the most important gains when visual perception. Representational similarity analysis (RSA) was leveraged to compare the brain features extracted by our model in Fig. 3(c). We could observe distinct intra-category aggregation, after grouping the fine-grained test concepts into four larger categories: animal, food, vehicle, and tool.

### 4.4.3 IMAGE GENERATION

To evaluate the visual decoding capability, we implemented an image generation pipeline using brain activity. Following Li et al. (2024); Benchetrit et al. (2023), we trained a diffusion prior to process the brain embeddings, then used pre-trained SDXL Podell et al. (2023) and IP-Adapter Ye et al. (2023)

Table 3: Metrics of image generation.

| | Model | PixCorr ↑ | SSIM ↑ | AlexNet(2) ↑ | AlexNet(5) ↑ | Inception ↑ | CLIP ↑ | EfficientNet ↓ | SwAV ↓ |
|---|---|---|---|---|---|---|---|---|---|
| | MB2C | 0.188 | 0.333 | - | - | - | - | - | - |
| EEG (sub-08) | ATM | 0.160 | 0.345 | **0.776** | 0.866 | 0.734 | 0.786 | - | 0.582 |
| | **BrainAE** | **0.211** | **0.432** | 0.768 | **0.869** | **0.753** | **0.816** | 0.865 | **0.541** |
| | B. D. | 0.081 | 0.341 | **0.788** | 0.879 | 0.710 | 0.799 | - | 0.560 |
| MEG (sub-02) | ATM | 0.104 | 0.340 | 0.613 | 0.672 | 0.619 | 0.603 | - | 0.651 |
| | **BrainAE** | **0.181** | **0.386** | 0.767 | **0.883** | **0.745** | **0.814** | 0.878 | **0.553** |
| **MUA** (average) | **BrainAE** | 0.221 | 0.424 | 0.856 | 0.931 | 0.799 | 0.839 | 0.834 | 0.539 |

Table 4: Ablation study for objective functions.

| | ENCODING | | | | DECODING | | | |
|---|---|---|---|---|---|---|---|---|
| | All channels | | Visual channels | | Classification | | Retrieval | |
| objective | MSE ↓ | Pearson's r ↑ | MSE ↓ | Pearson's r ↑ | Top-1 ↑ | Top-5 ↑ | Top-1 ↑ | Top-5 ↑ |
| w/ $L_{con}$ | 0.163±0.067 | 0.000±0.006 | 0.177±0.078 | -0.005±0.008 | 18.1±3.9 | 47.1±4.5 | **30.9**±6.0 | **63.5**±5.2 |
| w/ $L_{recon}$ | 0.027±0.005 | 0.469±0.105 | 0.027±0.005 | 0.575±0.117 | 0.6±0.5 | 3.3±0.6 | 0.7±0.3 | 3.1±0.5 |
| w/o $L_{con}$ | 0.033±0.007 | 0.537±0.073 | 0.028±0.005 | 0.655±0.057 | 1.5±0.5 | 6.8±1.1 | 1.3±0.7 | 9.2±2.3 |
| w/o $L_{fea}$ | 0.028±0.004 | 0.498±0.089 | 0.029±0.003 | 0.641±0.060 | 17.8±2.6 | 46.4±2.9 | 30.0±4.5 | 63.2±5.3 |
| w/o $L_{recon\_B}$ | 0.043±0.008 | 0.481±0.094 | 0.034±0.004 | 0.621±0.084 | 18.0±3.6 | 45.6±3.9 | 30.1±3.5 | 64.0±5.4 |
| w/o $L_{recon\_I}$ | 0.072±0.008 | 0.392±0.065 | 0.067±0.007 | 0.469±0.064 | 18.0±3.0 | 46.1±6.2 | 30.0±5.5 | 61.8±7.7 |
| overall | **0.027**±0.005 | **0.583**±0.071 | **0.024**±0.003 | **0.680**±0.052 | **18.2**±3.2 | **46.7**±4.3 | 30.6±4.0 | 63.0±4.8 |

for image generation. Example raw and generated images of the test set are shown in Fig. 3(d). We can see that the low-level structural information and the high-level semantic information have been recovered to a large margin. The evaluation metrics are given in Table 3, including PixCorr, SSIM, AlexNet, Inception, CLIP score, and SwAV. BrainAE shows competitive results with other works, such as ATM Li et al. (2024), MB2C Wei et al. (2024), and B.D. Benchetrit et al. (2023).

## 4.5 MODEL ANALYSIS

### 4.5.1 ENCODING AND DECODING CONSISTENCY

To demonstrate the effectiveness of the unified model and the consistency of bidirectional mapping, we first compare the correlation between decoding accuracy changes when masking different channels and the channel-level encoding Pearson's r. As shown in Fig. 3(e), the two tasks have high correlation with r=0.71. Similarly, the time-wise test, calculating between decoding accuracy changes when masking different time samples and the time-level encoding Pearson's r, also shows correlated results.

### 4.5.2 ABLATION STUDY

We perform an ablation study to show the impact of objective functions in Table 4. The Pearson's r of all channels and top-1 classification accuracy are treated as the primary indicators of encoding and decoding, separately. The $L_{recon}$ and $L_{con}$ assume a dominant role for encoding and decoding, because the correlation and accuracy achieve chance level when only using $L_{con}$ or $L_{recon}$. The Pearson's r value has a decrement of 0.191, 0.102, and 0.085 when training the model without $L_{reconI}$ ($p < 0.01$), $L_{reconB}$ ($p < 0.01$), and $L_{fea}$ ($p < 0.01$), respectively, while these objectives have no significant impact on the decoding performance ($p > 0.05$). On the other hand, $L_{con}$ significantly helps improve the top-1 accuracy by 16.7 ($p < 0.01$), and the Pearson's r by 0.046 ($p < 0.01$).

### 4.5.3 BACKBONE

We introduced well-designed feature extractors in BrainAE framework to evaluate the adaptability, including ShallowNet, DeepNet Schirrmeister et al. (2017), EEGNet Lawhern et al. (2018), and Conformer Song et al. (2023b) as the brain encoder, and pre-trained ViT-B/16 Dosovitskiy et al. (2021), CLIP-L/14, and CLIP-H/14 Radford et al. (2021) as the image encoder. Despite replacing TSConv and EVA-CLIP Sun et al. (2023), competitive results are achieved in encoding and decoding.

## 4.6 SIGNAL RECONSTRUCTION

It would be significant to explore neural mechanisms if complete brain activity could be predicted or reconstructed from partial recordings, in cases where complete recordings are not possible or under noise interference. Here, we set up an evaluation by masking the raw signals from random, spatial, temporal, and forecast ways, shown in Fig. 4(a), where the three EEG channels on the occipital area

Table 5: Brain and image encoder comparison

| | | ENCODING | | | | DECODING | | | |
|---|---|---|---|---|---|---|---|---|---|
| | | All channels | | Visual channels | | Classification | | Retrieval | |
| encoder | methods | MSE ↓ | Pearson's r ↑ | MSE ↓ | Pearson's r ↑ | Top-1 ↑ | Top-5 ↑ | Top-1 ↑ | Top-5 ↑ |
| BRAIN | DeepNet | 0.023±0.004 | 0.596±0.071 | 0.021±0.003 | 0.697±0.052 | 12.1±2.8 | 37.5±4.6 | 21.2±3.9 | 51.5±5.2 |
| | ShallowNet | 0.029±0.005 | 0.576±0.066 | 0.025±0.004 | 0.677±0.050 | 15.1±3.8 | 42.4±6.2 | 25.7±4.8 | 58.3±5.8 |
| | EEGNet | **0.023**±0.003 | **0.601**±0.067 | **0.020**±0.003 | **0.702**±0.047 | 16.4±2.9 | 41.5±3.3 | 27.4±3.7 | 58.3±5.3 |
| | Conformer | 0.025±0.004 | 0.589±0.065 | 0.022±0.003 | 0.690±0.050 | 17.2±3.8 | 44.1±5.9 | 26.7±4.2 | 58.7±6.1 |
| IMAGE | ViT-B/16 | 0.089±0.022 | 0.320±0.070 | 0.060±0.017 | 0.461±0.091 | 11.4±1.8 | 26.8±3.9 | 16.1±3.0 | 41.3±4.4 |
| | CLIP-L/14 | 0.024±0.004 | 0.572±0.072 | 0.023±0.003 | 0.667±0.064 | 14.0±2.2 | 41.1±5.5 | 19.3±4.2 | 50.4±6.8 |
| | CLIP-H/14 | 0.049±0.013 | 0.407±0.077 | 0.035±0.009 | 0.566±0.087 | 16.8±2.7 | 39.8±4.2 | 28.3±4.8 | 60.8±4.8 |
| | TSConv+EVA-CLIP | 0.027±0.005 | 0.583±0.071 | 0.024±0.003 | 0.680±0.052 | **18.2**±3.2 | **46.7**±4.3 | **30.6**±4.0 | **63.0**±4.8 |

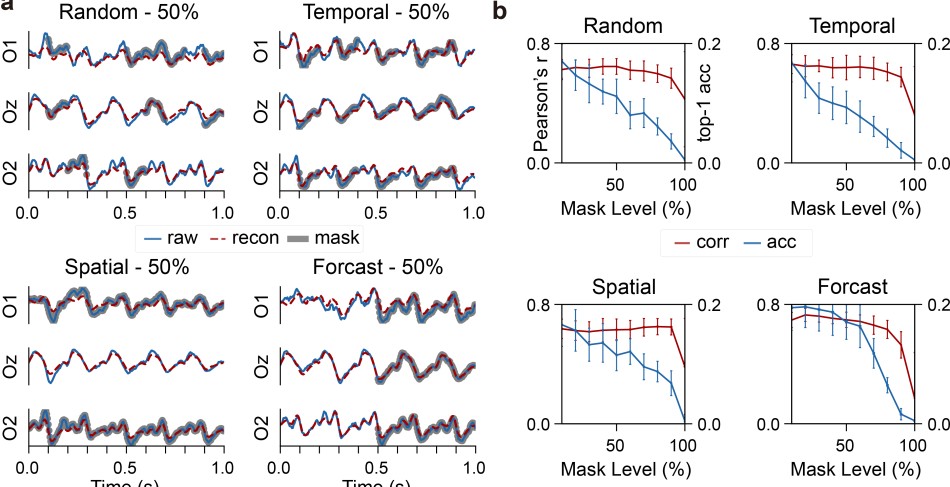

Figure 4: Brain activity reconstruction with four types of masked recordings. (a) The examples of three channels with random, spatial, temporal, and forecast masks at a ratio of 50%. (b) The encoding (in red) and decoding (in blue) performance under different masking types and levels.

of one trial are plotted. In implementation, the masked signals are processed by the brain encoder and then reconstructed by the brain decoder. The reconstructed signals align closely with the raw signals across all masking strategies, preserving amplitude and temporal patterns even under severe data loss.

We also complement the reconstruction quality reflected by Pearson's correlation and decoding top-1 accuracy with different mask levels in Fig. 4(b). When we increase the ratio, the signal is still maintained at a higher quality, but the information for image decoding gradually decreases.

## 5 DISCUSSION AND CONCLUSION

We present BrainAE, a framework that unifies visual encoding and decoding through a shared latent space aligned with visual features. By integrating both directions, BrainAE not only achieves strong predictive performance but also provides a computational tool for probing neural representations. Across EEG, MEG, and MUA datasets, the model achieves high correlations in encoding while preserving temporal, spatial, and frequency characteristics of neural activity. For decoding, BrainAE outperforms methods in image identification and achieves competitive performance in image generation. Beyond task metrics, our analyses demonstrate that BrainAE captures meaningful temporal, spatial, and semantic patterns, and exhibits consistency across encoding and decoding tasks. Its ability to reconstruct masked neural recordings further highlights the framework's robustness.

Despite its promising result, BrainAE has limitations that warrant further investigation. For instance, visual perception was chosen for evaluation, but other behaviors, such as motor and speech, need more extensive testing. Secondly, we chose electrophysiological recordings for comparison, due to the fast dynamics of visual processing. The generalization to other modalities, such as fMRI and fNIRS, remains unexplored. Future work may also explore BrainAE's applicability in BCI systems.

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

## A    ETHICS STATEMENT

This work uses previously published and publicly available human EEG/MEG datasets and macaque MUA datasets. All data were collected with informed consent (for human participants) or under approved animal care protocols, as stated by the original dataset providers. No new human or animal experiments were conducted in this study. Our framework is intended for advancing computational neuroscience and applications, not for clinical or invasive deployment. On the other hand, while BrainAE offers promising applications, decoding neural activity also raises potential concerns for privacy and misuse. Careful consideration of ethical safeguards, data consent, and responsible deployment is essential to ensure beneficial use.

## B    REPRODUCIBILITY STATEMENT

We have reported the details of our framework design and objective functions in Section 3, the model architecture in Appendix D.1, and the algorithm in Appendix E. We have also released the dataset details and preprocessing in Section 4 and Appendix F. The code will be made publicly available.

## C    THE USE OF LARGE LANGUAGE MODELS

We used large language models (LLMs) to assist with polishing the writing and improving readability. No part of the scientific content, analysis, or results was generated or influenced by LLMs. We thank the community for their development of many excellent LLMs to boost scientific communication.

# D MODEL DETAILS

## D.1 BRAIN ENCODER AND DECODER

Here, we first introduce the detailed architectures of the BrainAE framework, especially the brain encoder and brain decoder in implementation. As Table 6 mentioned, the brain encoder used temporal and spatial convolutional layers for spatial and temporal feature extraction.

We designed the brain decoder as in Table 7, mirroring the brain encoder with transposed convolution and an up-sampling layer. In the current version, the hyperparameters were set as follows: $k = 40$, $m_1 = 26$, $m_2 = 5$, and $s = 5$. This paper aims to verify the potential of the shared representations obtained by such an alignment-driven framework. We believe any further enhancement of the encoder and decoder architecture would help improve the overall performance.

Table 6: Architecture of the Brain Encoder.

| Layer | In | Out | Kernel | Stride | Dimension |
|---|---|---|---|---|---|
| Temporal Conv | 1 | $k$ | $(1, m_1)$ | $(1, 1)$ | $(b, k, C, T - m_1 + 1)$ |
| Avg Pooling | $k$ | $k$ | $(1, m_2)$ | $(1, s)$ | $(b, k, C, (T - m_1 - m_2 + 1)/s + 1)$ |
| Spatial Conv | $k$ | $k$ | $(ch, 1)$ | $(1, 1)$ | $(b, k, 1, (T - m_1 - m_2 + 1)/s + 1)$ |
| Flatten&Linear | | | | $[k * ((T - m_1 - m_2 + 1)/s + 1) \rightarrow$ dim of shared features$]$ | |

Table 7: Architecture of the Brain Decoder.

| Layer | In | Out | Kernel | Stride | Dimension |
|---|---|---|---|---|---|
| Linear&unFlatten | | | | dim of shared features$\rightarrow [k * ((T - m_1 - m_2 + 1)/s + 1)]$ | |
| Transposed Temporal Conv | 1 | $k$ | $(1, m_1)$ | $(1, 1)$ | $(b, k, C, T - m_1 + 1)$ |
| Upsampling | $k$ | $k$ | $(1, m_2)$ | $(1, s)$ | $(b, k, C, (T - m_1 - m_2 + 1)/s + 1)$ |
| Transposed Spatial Conv | $k$ | $k$ | $(ch, 1)$ | $(1, 1)$ | $(b, k, 1, (T - m_1 - m_2 + 1)/s + 1)$ |

## D.2 IMAGE ENCODER AND DECODER

We applied several large pre-trained image models as the image encoder to get image embeddings, such as: (i) ViT-B/16 Dosovitskiy et al. (2021) with 12 layers pre-trained on ImageNet-21k Ridnik et al. (2021) and finetuned on ImageNet 2012 Deng et al. (2009), (ii) CLIP-L/14 Radford et al. (2021) with 24 layers pre-trained on LAION-400M, (iii) CLIP-H/14 with 32 layers pre-trained on LAION-2B Schuhmann et al. (2022), and (iv) EVA-CLIP with 64 layers pre-trained on LAION-2B. In the implementation of the Image Encoder of BrainAE, we only add one linear layer trained to transfer the image embeddings to a shared space with brain embeddings.

After training, we could directly use the model for classification and retrieval by template matching with prepared templates. We also test the performance of image generation in two-stage ways, following the Ramesh et al. (2022); Scotti et al. (2023); Benchetrit et al. (2023); Li et al. (2024). The image generation process is formulated as follows:

$$P(\boldsymbol{X}_i^{IR}|\boldsymbol{z}_i^B) = P(\boldsymbol{X}_i^{IR}, \boldsymbol{z}_i^{I'}|\boldsymbol{z}_i^B) = P(\boldsymbol{X}_i^{IR}|\boldsymbol{z}_i^{I'}, \boldsymbol{z}_i^B)P(\boldsymbol{z}_i^{I'}|\boldsymbol{z}_i^B) \tag{6}$$

where we trained a U-Net-based prior model $P(\boldsymbol{z}_i^{I'}|\boldsymbol{z}_i^B)$ to transfer the model to transfer the brain embeddings $\boldsymbol{z}_i^B$ to the CLIP-space embeddings $\boldsymbol{z}_i^{I'}$, which is suitable for the image decoder $P(\boldsymbol{X}_i^{IR}|\boldsymbol{z}_i^{I'}, \boldsymbol{z}_i^B)$, pre-trained stable diffusion model. The SDXL model Podell et al. (2023) and IP-Adapter Ye et al. (2023) were used in the implementation. Note that we focus on validating BrainAE's latent representations, which can be equipped with other image-generation pipelines.

# E  ALGORITHM

The algorithm flow of BrainAE training processing is shown in Algorithm 1.

---

**Algorithm 1** Training Process of the BrainAE Framework

---

1: **Input:** Training brain activity $X_{train}^B$ and stimulus images $X_{train}^I$; randomly divided validation brain activity $X_{val}^B$ and stimulus images $X_{val}^I$.
2: **Model:** Brain encoder $\mathcal{E}_B$, brain decoder $\mathcal{D}_B$, image encoder $\mathcal{E}_I$, and image decoder $\mathcal{D}_I$.
3: Initialize model parameters and hyperparameters.
4: Initialize $best\_val\_loss = +\infty$.
5: **for** epoch $= 0$ to $ep\text{-}1$ **do**
6:    # Training phase
7:    **for** each batch in training data **do**
8:       # Extract features
9:       $\boldsymbol{z_i^B} = \mathcal{E}_B(X_i^B)$; $\boldsymbol{z_i^I} = \mathcal{E}_I(X_i^I)$
10:      # Reconstruct activity signals
11:      $\boldsymbol{X_i^{B'I}} = \mathcal{D}_B(\boldsymbol{z_i^I})$; $\boldsymbol{X_i^{B'B}} = \mathcal{D}_B(\boldsymbol{z_i^B})$
12:      # Compute loss functions (eq. (1), (2), (5))
13:      $loss = loss_{con} + loss_{fea} + loss_{recon}$
14:      # Update parameters of $\mathcal{E}_B$, $ji$, and $\mathcal{E}_I$.
15:   **end for**
16:   # Validation phase
17:   Compute $val\_loss$ using $\mathcal{E}_B$, $\mathcal{D}_B$, $\mathcal{E}_I$, on $X_{val}^B$, $X_{val}^I$.
18:   **if** $val\_loss < best\_val\_loss$ **then**
19:      Save the best checkpoint.
20:      $best\_val\_loss = val\_loss$
21:   **end if**
22: **end for**

---

# F  DATASETS

We summarize the details of the three datasets used for comparative experiments in Table 8. It gives the number of subjects, recording channels, training and testing set sizes, and stimulus onset asynchrony (SOA), covering three types of brain recordings: EEG, MEG, and Spike data.

(i) THINGS-EEG2 Gifford et al. (2022) consists of EEG recordings from 10 subjects with 63 channels. The training set includes 1,654 concepts across 10 conditions, each repeated 4 times, while the test set contains 200 concepts with a single condition and 80 repetitions. The SOA is 200 ms with a 100 ms stimulation window.

(ii) THINGS-MEG Hebart et al. (2023) involves MEG data from 4 subjects with 271 channels. The training set includes 1,854 concepts (minus 200 for validation), each with 12 conditions and 1 repetition. The test set has 200 concepts with a single condition and 12 repetitions. The SOA varies around 1,500 ms ± 200 ms, with a 500 ms stimulation window.

(iii) TVSD Papale et al. (2025) includes Spike recordings from 2 subjects with 1,024 channels. The training set consists of 1,854 concepts (minus 100 for validation), each with 12 conditions and 1 repetition. The test set contains 100 concepts, each with 1 condition and 30 repetitions. The SOA is 400 ms with a 200 ms stimulation window.

Table 8: Datasets for comparative experiments.

|  | Type | Subjects | Channels | Train* | Test* | SOA (stimulation) |
|---|---|---|---|---|---|---|
| THINGS-EEG2 | EEG | 10 | 63 | 1654 \| 10 \| 4 | 200 \| 1 \| 80 | 200 (100) ms |
| THINGS-MEG | MEG | 4 | 271 | (1854-200) \| 12 \| 1 | 200 \| 1 \| 12 | 1500±200 (500) ms |
| TVSD | Spike | 2 | 1024 | (1854-100) \| 12 \| 1 | 100 \| 1 \| 30 | 400 (200) ms |

* concepts (classes) | conditions (images) | repetitions (times).

# G ENCODING RESULTS

We reported the final results of each dataset after running the model 5 times with different random seeds. The encoding performance of each human or macaque participant is provided in Table 9, 10, 11, respectively. The performance is evaluated using MSE, where lower is better, and Pearson's correlation coefficient, where higher is better, across multiple subjects. BrainAE outperforms linear models Gifford et al. (2022) in three datasets, achieving lower MSE and higher correlation.

Table 9: Overall encoding performance on EEG dataset (N=10).

| Method | Sub 1 | | Sub 2 | | Sub 3 | | Sub 4 | | Sub 5 | | Sub 6 | | Sub 7 | | Sub 8 | | Sub 9 | | Sub 10 | | Ave | |
|---|---|---|---|---|---|---|---|---|---|---|---|---|---|---|---|---|---|---|---|---|---|---|
| | MSE ↓ | r ↑ | MSE | r | MSE | r | MSE | r | MSE | r | MSE | r | MSE | r | MSE | r | MSE | r | MSE | r | MSE | r |
| Linear (all ch) | 0.033 | 0.522 | 0.043 | 0.479 | 0.030 | 0.518 | 0.042 | 0.469 | 0.044 | 0.395 | 0.046 | 0.378 | 0.034 | 0.514 | 0.033 | 0.611 | 0.030 | 0.380 | 0.037 | 0.533 | 0.037 | 0.480 |
| Linear (visual ch) | 0.029 | 0.664 | 0.036 | 0.519 | 0.028 | 0.631 | 0.034 | 0.584 | 0.040 | 0.484 | 0.041 | 0.526 | 0.030 | 0.668 | 0.030 | 0.647 | 0.026 | 0.515 | 0.031 | 0.658 | 0.033 | 0.590 |
| **BrainAE** (all ch) | 0.024 | 0.627 | 0.032 | 0.602 | 0.020 | 0.633 | 0.03 | 0.588 | 0.033 | 0.500 | 0.035 | 0.465 | 0.025 | 0.618 | 0.024 | 0.696 | 0.021 | 0.484 | 0.027 | 0.620 | **0.027** | **0.583** |
| **BrainAE** (visual ch) | 0.021 | 0.744 | 0.027 | 0.633 | 0.020 | 0.720 | 0.024 | 0.686 | 0.028 | 0.604 | 0.030 | 0.624 | 0.023 | 0.738 | 0.023 | 0.717 | 0.018 | 0.619 | 0.025 | 0.716 | **0.024** | **0.680** |

Table 10: Overall encoding performance on MEG dataset (N=4).

| Method | Sub 1 | | Sub 2 | | Sub 3 | | Sub 4 | | Ave | |
|---|---|---|---|---|---|---|---|---|---|---|
| | MSE ↓ | r ↑ | MSE | r | MSE | r | MSE | r | MSE | r |
| Linear (all ch) | 0.421 | 0.460 | 0.684 | 0.575 | 1.237 | 0.396 | 0.482 | 0.353 | 0.706 | 0.446 |
| Linear (visual ch) | 0.410 | 0.527 | 0.742 | 0.661 | 0.356 | 0.408 | 0.509 | 0.435 | 0.754 | 0.508 |
| **BrainAE** (all ch) | 0.373 | 0.511 | 0.620 | 0.614 | 1.079 | 0.455 | 0.426 | 0.391 | **0.624** | **0.493** |
| **BrainAE** (visual ch) | 0.366 | 0.571 | 0.678 | 0.692 | 1.181 | 0.470 | 0.450 | 0.472 | **0.669** | **0.551** |

Table 11: Overall encoding performance on Spike dataset (N=2).

| Method | Sub 1 | | Sub 2 | | Ave | |
|---|---|---|---|---|---|---|
| | MSE ↓ | r ↑ | MSE | r | MSE | r |
| Linear (all ch) | 0.140 | 0.755 | 0.143 | 0.836 | 0.141 | 0.796 |
| Linear (visual ch) | 0.070 | 0.586 | 0.087 | 0.792 | 0.079 | 0.689 |
| **BrainAE** (all ch) | 0.310 | 0.853 | 0.274 | 0.852 | **0.292** | **0.852** |
| **BrainAE** (visual ch) | 0.019 | 0.938 | 0.113 | 0.859 | **0.066** | **0.898** |

# H    DECODING RESULTS

Similarly, the decoding performance of each participant is provided in Table 12, 13, 14, respectively. The performance of classification and retrieval tasks is evaluated using top-1 and top-5 accuracy across multiple subjects. The results from BraVL Du et al. (2023), NICE Song et al. (2023a), ATM Li et al. (2024), and MB2C Wei et al. (2024) are invovlved for comparison. Note that since the authors of ATM did not provide results for individual subjects, the results in the below table were reproduced, and its top-1 acc 28.5% and top-5 acc 60.4% were close to that mentioned in the original article, where the top-1 acc was 28.6% and the top-5 acc was 58.5%. BrainAE achieves higher results than other models across all datasets.

Table 12: Overall decoding performance on EEG dataset (N=10).

| Method | Sub 1 | | Sub 2 | | Sub 3 | | Sub 4 | | Sub 5 | | Sub 6 | | Sub 7 | | Sub 8 | | Sub 9 | | Sub 10 | | Ave | |
|---|---|---|---|---|---|---|---|---|---|---|---|---|---|---|---|---|---|---|---|---|---|---|
| | top-1 ↑ | top-5 ↑ | top-1 | top-5 | top-1 | top-5 | top-1 | top-5 | top-1 | top-5 | top-1 | top-5 | top-1 | top-5 | top-1 | top-5 | top-1 | top-5 | top-1 | top-5 | top-1 | top-5 |
| Classification | | | | | | | | | | | | | | | | | | | | | | |
| BraVL Du et al. (2023) | 6.1 | 17.9 | 4.9 | 14.9 | 5.6 | 17.4 | 5.0 | 15.1 | 4.0 | 13.4 | 6.0 | 18.2 | 6.5 | 20.4 | 8.8 | 23.7 | 4.3 | 14.0 | 7.0 | 19.7 | 5.8 | 17.5 |
| NICE Song et al. (2023a) | 12.3 | 36.6 | 10.4 | 33.9 | 13.1 | 39.0 | 16.4 | 47.0 | 8.0 | 26.9 | 14.1 | 40.6 | 15.2 | 42.1 | 20.0 | 49.9 | 13.3 | 37.1 | 14.9 | 41.9 | 13.8 | 39.5 |
| **BrainAE** | 18.6 | 44.2 | 15.7 | 40.7 | 18.2 | 47.5 | 22.6 | 51.5 | 11.9 | 39.0 | 18.5 | 48.2 | 19.3 | 48.4 | 23.2 | 53.4 | 15.1 | 45.1 | 18.4 | 48.6 | **18.2** | **46.7** |
| Retrieval | | | | | | | | | | | | | | | | | | | | | | |
| ATM | 20.5 | 58.0 | 18.0 | 47.5 | 25.0 | 60.0 | 27.5 | 58.0 | 15.5 | 42.0 | 27.5 | 63.5 | 24.0 | 53.0 | 41.0 | 72.0 | 21.5 | 51.0 | 36.5 | 69.5 | 25.7 | 57.5 |
| MB2C | 23.7 | 56.3 | 22.7 | 50.5 | 26.3 | 60.2 | 34.8 | 67.0 | 21.3 | 53.0 | 31.0 | 62.3 | 25.0 | 54.8 | 39.0 | 69.3 | 27.5 | 59.3 | 33.2 | 70.8 | 28.5 | 60.4 |
| **BrainAE** | 27.2 | 57.3 | 27.6 | 59.1 | 31.5 | 65.6 | 36.0 | 71.0 | 26.3 | 54.8 | 32.3 | 63.0 | 26.1 | 63.1 | 38.9 | 69.5 | 30.6 | 63.3 | 29.2 | 63.6 | **30.6** | **63.0** |

Table 13: Overall decoding performance on MEG dataset (N=4).

| Method | Sub 1 | | Sub 2 | | Sub 3 | | Sub 4 | | Ave | |
|---|---|---|---|---|---|---|---|---|---|---|
| | top-1 ↑ | top-5 ↑ | top-1 | top-5 | top-1 | top-5 | top-1 | top-5 | top-1 | top-5 |
| Classification | | | | | | | | | | |
| NICE | 6.9 | 20.5 | 15.3 | 37.1 | 12.3 | 35.0 | 5.8 | 21.1 | 10.1 | 28.4 |
| **BrainAE** | 10.0 | 25.4 | 20.2 | 45.9 | 17.1 | 43.6 | 9.8 | 25.3 | **14.3** | **35.1** |
| Retrieval | | | | | | | | | | |
| NICE | 9.6 | 27.8 | 18.5 | 47.8 | 14.2 | 41.6 | 9.0 | 26.6 | 12.8 | 36.0 |
| ATM | 11.5 | 32.0 | 29.0 | 65.5 | 24.0 | 48.5 | 9.0 | 30.5 | 18.4 | 44.1 |
| **BrainAE** | 12.9 | 35.0 | 33.1 | 65.7 | 26.4 | 57.1 | 13.2 | 35.5 | **21.4** | **48.3** |

Table 14: Overall decoding performance of BrainAE on Spike dataset (N=2).

| Method | Sub 1 | | Sub 2 | | Ave | |
|---|---|---|---|---|---|---|
| | top-1 ↑ | top-5 ↑ | top-1 | top-5 | top-1 | top-5 |
| Classification | 28.8 | 65.6 | 25.0 | 58.4 | **26.9** | **62.0** |
| Retrieval | 49.6 | 81.2 | 37.4 | 72.8 | **43.5** | **77.0** |

## I  PARAMETERS

We set the temperature parameter $\tau$ to 0.07 following Radford et al. (2021). Below, we provide additional comparisons in Table 15, where the 0.01-level shows better performance. The temperature has a greater impact on decoding than encoding. The r-value with $\tau = 0.07$ is slightly higher than with $\tau = 0.7$ and $\tau = 0.007$ (p > 0.05), while the accuracy with $\tau = 0.07$ is significantly higher than those (p < 0.01).

Table 15: Comparison of temperature parameter with EEG dataset.

| $\tau$ | Pearson's r (all ch) | top-1 acc (classification) |
|---|---|---|
| 0.3 | 0.573±0.068 | 15.8±3.3 |
| 0.3 | 0.576±0.070 | 14.6±3.0 |
| 0.7 | 0.578±0.071 | 15.0±3.3 |
| 0.03 | 0.593±0.065 | 19.4±3.6 |
| 0.05 | 0.590±0.064 | 19.5±3.7 |
| 0.07 | 0.583±0.066 | 19.2±2.7 |
| 0.003 | 0.564±0.075 | 13.1±2.0 |
| 0.005 | 0.569±0.071 | 13.9±3.2 |
| 0.007 | 0.579±0.075 | 14.8±2.8 |

## J  COMPUTATIONAL COST

To evaluate the usability in real BCI scenarios, we report the coarse computational time to train the model for each recording on one GeForse 4090 GPU, as shown in Table 15.

Table 16: Computational time on one GPU.

| 1× GPU | training time per subject | test time each trial |
|---|---|---|
| EEG | 7 min | 4.7e-5 s |
| MEG | 19 min | 9.5e-5 s |
| MUA | 28 min | 5.7e-4 s |

## K  BROADER IMPACT

BrainAE unifies visual encoding and decoding within a bidirectional latent space, providing a new computational tool for studying neural representations and the mechanisms of visual processing. By bridging neuroscience and machine learning, it contributes to advancing both our understanding of biological intelligence and the design of brain-inspired AI systems. Beyond research, the framework's robustness and efficiency make it promising for real-world applications such as brain–computer interfaces, assistive technologies, and cognitive state monitoring.

