# OpenReview forum: "BrainAE: Alignment-driven Autoencoder for Bidirectional Visual Encoding and Decoding"
_ICLR.cc/2026/Conference — Submitted to ICLR 2026_

### Official Review · Reviewer_sswh · 2025-10-15

**Soundness:** 3
**Presentation:** 3
**Contribution:** 2
**Rating:** 4
**Confidence:** 4

**Summary:**

the paper describes an encoder/decoder system to enable a bidirectional dialogue between visual stimuli and corresponding neural activity. The paper is mainly oriented on the technical description of the system that has been built and its evaluation, through the selection of well-chosen benchmarks.

**Strengths:**

The approach is solid and well described. The state of the art is also of good quality. Figure 1 is clear and describes well the method.

**Weaknesses:**

Nevertheless, in spite of this globally good quality, I found the approach not very innovative. In addition, it is not very discussed *why* developing such a model could be interesting, its impact and limitations.

**Questions:**

Could you clearly state why it would be interesting to have such a tool at disposal and in which domain(s) and for which reason the impact would be important ?

---

### Official Review · Reviewer_wiW8 · 2025-10-30

**Soundness:** 2
**Presentation:** 1
**Contribution:** 2
**Rating:** 2
**Confidence:** 3

**Summary:**

The paper introduces BrainAE, an autoencoder-based model that is trained to reconstruct raw electrophysiology signals while aligning its latent space to the one of a visual embedding. This autoencoder can be used simultaneously for the task of visual encoding, visual decoding, and reconstructing masked brain activity (temporally or spatially). This bidirectional approach supports both visual encoding and decoding, which are often considered as separate tasks. It is evaluated on three datasets of Human EEG, Human MEG, and MUA, on visual encoding task (MSE, Pearson’s r), visual decoding task (template-based classification accuracy, retrieval top-1/top-5 accuracy, generation using image similarity metrics), and brain activity reconstruction. The paper shows superior correlations on encoding compared to a linear baseline and superior decoding performance on MEG / EEG compared to previous baselines.

**Strengths:**

- The approach is simple: a convolutional autoencoder, MSE and CLIP losses at raw and latent level.
- The motivation for this approach is clearly stated (merging encoding and decoding into a single task) and the presented results correspond to the claims in the abstract.
- The Methods section is clearly written and can be easily followed

**Weaknesses:**

- Several important sections of the paper are difficult to follow and lack details to be understandable (see questions)
- Some important details are missing (e.g. the dimensionality of the latent space of the AE, training of the diffusion prior, which classes are used as templates and if they are the same as in with previous works, ...)
- Some scores are reported only on one specific subject from the dataset and the reason is not discussed (e.g. Image similarity metrics on reconstruction task on MEG dataset is reported only for subject 2)
- Image reconstructions shown are of SOTA quality compared to previous works but this is not discussed further, either the paper should include a broader / more faithful view of the quality of these reconstructions or argue that this reconstruction quality is consistent over all datasets.

**Questions:**

- Section 4.3.1 on encoding refers to an ‘alignment’ process used for encoding, which is not defined prior to this section and compares the results of encoding with two ‘Linear Dec’ and ‘Nonlinear Dec’ settings which are not defined either. These processes are named ‘decoders’, which makes it even more confusing in the context of an encoding task. Since in the ‘Problem definition’ section, the encoding task is described as ‘Visual encoding, in Fig. 1(b), gets image embeddings by the encoder E_I and input to the decoder D_B for the encoded brain activity’, it is not clear whether this is the process actually done for evaluating encoding or if it’s something else involving an alignment module.
- Figure 3. The companion text for panel A refers to top-1 accuracy changes with pointwise masking but the y-axis contains negative values (incompatible with an accuracy metric), is this plot actually showing an accuracy metric on the y-axis or something else whose value can be negative ? It is unclear what the error bars in panel B correspond to.
- It would be great to have a clearer view of the quality of reconstructions obtained from SDXL, by sharing not only the best cases but some worse and failure cases.
- The reconstruction process is not detailed: it is not clear how the training of the diffusion prior for mapping the image encoder embedding to an SDXL prompt embedding is done, nor how the IP-adapter is used.
- The conclusion of Backbone section 4.5.3 is not clear: “Despite replacing TSConv and EVA-CLIP Sun et al. (2023), competitive results are achieved in encoding and decoding”. Does it mean that TSConv and EVA-CLIP provide better results than other models?
- What time / space subsets are removed for obtaining Fig 3e ?
- It is not clear what the dimension of the AE is, and whether ablations have been carried out on this dimension. It is implicitly assumed that it is the same as the output of the image encoder, but confusing because a linear layer is used to project the image encoder output to the AE latent. This should be clarified so we know whether the AE latent has the same dimension as the image encoder output or whether it differs, and if so, provide ablations of the chosen dimensionality.

---

### Official Review · Reviewer_z5xD · 2025-11-01

**Soundness:** 2
**Presentation:** 2
**Contribution:** 2
**Rating:** 2
**Confidence:** 4

**Summary:**

The paper proposes BrainAE, which aligns brain activity embeddings (from a temporal–spatial conv encoder/decoder) with frozen image embeddings (e.g., CLIP/EVA-CLIP) in a shared latent via a combination of InfoNCE, latent MSE (brain–image), and brain-signal reconstruction losses. Downstream, the model supports (a) image→brain “encoding,” (b) brain→image “decoding” via retrieval or a separate diffusion pipeline, and (c) masked brain-signal reconstruction. Experiments cover EEG/MEG and macaque MUA.

**Strengths:**

1. BrainAE is a simple, unified latent for both encoding and decoding is easy to reproduce and reason about.
2. Broad data coverage (EEG/MEG/MUA) is welcome.
3. Some loss ablations are included.

**Weaknesses:**

**Methodological issues**

1. It's not truly “bidirectional autoencoding.” The image side is frozen; there is no jointly trained image decoder. Image generation uses a separate diffusion prior conditioned on brain→CLIP. As implemented, this is an alignment-regularized brain autoencoder + frozen image encoder, not a symmetric/bidirectional AE.

2. Contrastive setup under-specified. How are negatives formed? pure in-batch? Are batches mixed across subjects/classes? Any hard-negative mining? These choices crucially affect alignment strength and the attainable upper bound.

3. (Important part) Baselines are weak/unclear. The paper cites a “linear decoder,” but the exact form is not specified. Also, strong, simple, more up-to-date, must-have baselines are missing (e.g. CLIP feature -> brain)


**Experimental issues**

1. Comparisons to recent work are incomplete. The paper does not line up with several strong recent methods, and the ATM classification scores reported here are far below the same metric reported elsewhere. This strongly suggests protocol/implementation mismatches (splits, zero-shot setup, template construction, backbone versions, etc.). Authors must align evaluation protocols.

2. Template-based decoding is fragile and under-controlled. From where are the templates sampled? How many per class? Are they fixed across runs? Report the template lists, ensure zero-shot semantics where claimed, and show variance across random template sets.

3. RSVP overlap confound (EEG). With ~200 ms SOA, single-trial epochs contain overlapping responses to multiple items. Without deconvolution/overlap correction, “image-specific” decoding claims are weakened. Either deconvolve or replicate the temporal-importance analysis in MEG (longer SOA).

4. Motivation for visual encoding is under-argued. The paper should justify practical benefits (e.g., as a self-supervised signal that improves decoding, as in-silico stimulus design for closed-loop, or for data augmentation). No such downstream gain is currently demonstrated.

5. Presentation. Citation style is inconsistent; several notation/typo issues remain.

**Questions:**

1. Please add experimental details (e.g. loss weights, latent dimension, normalization, and batch construction (subject/class mixing).)
2. Please strengthen baselines to ensure apples-to-apples comparisons.

---

### Meta-Review · Area_Chair_VrXm · 2026-01-09

**Summary:**

This submission presents BrainAE, a spatiotemporal convolutional autoencoder for electrophysiology that is trained to reconstruct brain signals while aligning its latent to a frozen visual embedding space using a mix of reconstruction, latent MSE, and contrastive (InfoNCE) losses. The paper targets a unified framework for encoding and decoding, and conducts experiments across EEG, MEG, and macaque MUA. Reviewers appreciated the simplicity= of the idea and the breadth of datasets, and that some ablations are provided.

However, the reviews identified substantial issues in positioning, experimental clarity, and evaluation rigor. Reviewer z5xD note weak or unclear baselines and missing “must-have” comparisons (including stronger and more up-to-date CLIP-related baselines), incomplete alignment with recent work and reported protocol mismatches (splits, zero-shot/template construction, backbone versions), and fragile template-based decoding without adequate controls or variance reporting across template sets. Reviewer z5xD expressed concerns that the “bidirectional autoencoding” framing appears overstated: the image side is frozen and image generation relies on an external diffusion pipeline, making the approach closer to an alignment-regularized brain autoencoder plus a frozen image encoder rather than a symmetric/bidirectional autoencoder. Multiple reviewers (wiW8, z5xD) note key technical details are underspecified or confusing. Reviewer z5xD notes that the RSVP setting raises an overlap confound at short SOAs that is not addressed via deconvolution or corroborating analyses in conditions less susceptible to overlap.

While the goal of a unified latent supporting both encoding- and decoding-style tasks is interesting, the current manuscript does not provide sufficient evidence needed to support its claims. The issues raised by the reviewers remain unaddressed due to the lack of a rebuttal from the authors. Therefore, the manuscript is not yet ready for publication.

**Reviewer Concerns:**

The authors did not submit a rebuttal and thus all of the reviewer concerns remain.

**Reviewer Scores:**

The scores would stay the same.

---

### Decision · Program_Chairs · 2026-01-26

Reject